# Spatial Differences in Diet Quality and Economic Vulnerability to Food Insecurity in Bangladesh: Results from the 2016 Household Income and Expenditure Survey

**Mst. Maxim Parvin Mitu** [1,2]**, Khaleda Islam** [1]**, Sneha Sarwar** [1]**, Masum Ali** [3] **and Md. Ruhul Amin** [1,*]

1   Institute of Nutrition and Food Sciences, University of Dhaka, Dhaka 1000, Bangladesh;
    maximparvinmitu@gmail.com (M.M.P.M.); aruna@du.ac.bd (K.I.); snehasarwar4@gmail.com (S.S.)
2   Directorate General of Health Services, Mohakhali, Dhaka 1212, Bangladesh
3   International Food Policy Research Institute (IFPRI), Dhaka 1212, Bangladesh; masum.ali@cgiar.org
*   Correspondence: ruhul.infs@du.ac.bd

**Abstract:** The study explored the spatial differences in diet quality and economic vulnerability to food insecurity with the association of sociodemographic characteristics at the household level in Bangladesh. This study was a secondary data analysis of Household Income and Expenditure Survey (HIES) data of 2016. Both statistical and spatial analyses were applied while assessing diet qualities in terms of the household dietary diversity score (HDDS), percentage of food energy from staples (PFES), and percentage of expenditure on food (PEF) as an indicator of the economic vulnerability to food insecurity (EVFI). The study's findings revealed that the quality of people's diets worsened as they moved from urban to rural area, and EVFI increased as they moved from the center to the periphery of the country. Nationally, the average HDDS was about 6.3, and the average PFES per household per day was about 70.4%. The spatial distribution of HDDS and PFES showed that rural regions in terms of settlements and the north, northwest and southeast regions had mostly low diet diversity. Besides, the average PEF per household per day was about 54%, with the highest in Mymensingh (57.4%) and the lowest in Dhaka division (50.2%). The average PEF in the households illustrated was highest in rural (55.2%) and lowest in the city (45.7%). Overall, based on the PEF at the sub-district level, the medium level of vulnerability comprised the highest share (69%) in Bangladesh. Age, gender, literacy, educational qualification and religion of the household's head along with the number of earners, monthly income, area of settlements and divisions were significantly correlated with HDDS, PFES and EVFI. The study findings suggest that targeted interventions, including access to education, women empowerment and employment generation programs should be implemented in peripheral areas (north, northwest and southeast) to increase diet quality and minimize economic vulnerability to achieve sustainable food and nutrition security in Bangladesh.

**Keywords:** diet quality; economic vulnerability; food insecurity; Household Income and Expenditure Survey; Bangladesh

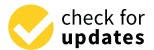



## 1. Introduction

Food insecurity is one of the major development problems in the world [1] which can predispose the entire population to malnutrition. Food insecurity is referred to as having limited or unpredictable physical and economic access to secure, adequate and healthy food to fulfill one's dietary needs or preferences [2]. A healthy, balanced and nutrient-dense diet is one of the important factors influencing health, productivity and survival, and it also has the potential to combat chronic diseases [3,4]. In resource-poor settings throughout the world, it is becoming increasingly acknowledged that low food quality, rather than inadequate energy consumption, is the fundamental nutritional constraint [1]. Poor and nutritionally deficient diets were responsible for roughly 22% of all deaths and 15% of all disability-adjusted life years (DALYs) among people worldwide [5]. Hence, it is important

to determine food security status as well as access to quality food in the nation to ensure a healthy life.

Several different factors affect food security status; the economy is one of the major role players. Research has also linked total food-related expenditure to food security and diet quality [6–8]. According to a recent study, households that spend a significant portion of their income on food, irrespective of their current state of consumption, are prone to economic vulnerability to food insecurity (EVFI) [9]. The logic behind this is straightforward. Any disruption in income might translate into a reduction in diet quality and overall food consumption, hence food insecurity [1]. Even though households can be food secure by expending the majority of their earnings on food, that cannot ensure a quality diet. The quality of a diet depends on access to diverse food groups, as this determines both energy and nutrient sufficiency. Traditionally, access to rice has been considered a crucial constituent of food security, as rice and wheat account for about three-quarters of the total energy intake and nearly half of all food-related expenditure in Bangladesh [8]. Nevertheless, major dependence on staples can exhibit the consumption of a poorer diet [1].

Another major factor in defining food security is the area of settlement. People around different geo locations tend to adhere to different dietary patterns. The Behavioral Risk Factor Surveillance System (BRFSS) has shown that the prevalence of different metabolic disorders, such as overweight, obesity and type 2 diabetes, is increasing regionally and racially in the US population [10]. According to BRFSS, the south has the greatest mortality and morbidity risk, followed by the middle west, north east and north [10]. When serum nutrient level was studied, the same trend was observed, where poor concentration of nutrients was observed in those residing in the south compared to the north [11]. Again, when the quality of their food was analyzed, it was determined that people in the south/middle east consumed a more energy-dense, nutrient-poor diet compared to the north/northwest population [12]. Three of these studies demonstrate how regional differences in food quality led to regional differences in blood nutrient concentrations, placing one part of the nation at a greater risk of illness than the other. Thus, spatial exploration can provide basic diagrammatic insight into dietary habits beyond conventional approaches within a certain region [13]. Even unequal distribution of wealth and living status of a household could play a vital role in the socioeconomic inequalities of child malnutrition, as well as food insecurity [14]. Thus, the role of a living place should be underscored in determining the quality of food consumed and the economic vulnerability to food insecurity.

To understand sustainable food security and sustainable diet, an emphasis on regional factors is critical. Currently, more than enough food is produced to support the world's population, yet the issue of food insecurity remains, with substantial disparities across nations and even within the same country, as an unsustainable food system leads to food insecurity and malnutrition [15]. The present agricultural and food system, often known as the industrialized system, places significant strain on ecosystems, resulting in food scarcity and nutritional deficiencies [16]. To achieve a sustainable food security system with adequate diet quality, the consideration of the regional situation is thus vital. This understanding can assist policymakers to implement targeted strategies in geographically vulnerable communities to assure food nutrient adequacy, affordability and availability, ultimately enhancing food sustainability and human nutrition outcomes [17].

By increasing national rice production, modernizing the infrastructure and liberalizing agricultural input and output markets, Bangladesh has made tremendous headway in addressing food security [18]. To create a long-term food security system, the whole country must have the same level of food security. There are limited data to determine whether national agriculture and food systems in Bangladesh are delivering nutritious foods or whether people are following the recommended dietary guidelines. It is critical to examine the regional condition to ensure a sustainable food security system throughout the country. To date, no study has been conducted considering the spatial difference at the sub-national level as the main outcome of diet quality and economic vulnerability to

food insecurity and their associated factors with it. However, there are a limited number of studies conducted elsewhere using large geographical units (i.e., rural/urban type or by administrative division or districts) focusing on food insecurity or poverty [19–21]. None of these studies included the smallest geographical units, such as the sub-district level (Upazila level) in their analyses. Among the various tools, many studies have used only household dietary diversity score (HDDS) to assess dietary diversity and micronutrient adequacy [8,22]. However, neither do they use multiple indicators of diet quality (HDDS and PFES) nor do they attempt to study diet quality and EVFI together. As a result, nationally representative data on the quality of food consumed and EVFI differences are scarce. Given this reality, this study aimed to investigate the regional differences in diet quality and EVFI among Bangladesh's entire population.

## 2. Methods

### 2.1. Data Source and Study Population

The study was conducted using a secondary analysis of food consumption data from the 2016 Household Income and Expenditure Survey (HIES). HIES is a nationwide survey conducted by the Bangladesh Bureau of Statistics (BBS) using a two-stage stratified random sampling technique [23]. The detailed methodology of the HIES-2016 survey has been described elsewhere [23]. The HIES 2016 observed a total of 46,080 households; however, four families were excluded from the study due to insufficient data. Finally, a total of 46,076 households were included in the study.

### 2.2. Food Data

To collect food consumption data, HIES 2016 consists of a distinct part with 145 categories of food items during the preceding 14 days using seven 2-day diaries [24]. A total of 125 food items of HIES data were considered in the diet quality analysis, excluding cigarettes and other smoking items.

### 2.3. Data Processing and Management

In this study, diet quality was measured in terms of household dietary diversity and the percentage of food energy from staples [1].

#### 2.3.1. Household Dietary Diversity Score (HDDS)

HDDS can be defined as the number of distinct food groups consumed per household per day. It is calculated in 12-point scores, and it indicates the ability of household economic access to foods [25]. In this study, HDDS was calculated for 14 days, and the total HDDS was divided by 14 to estimate HDDS per day. The HDDS is then divided into tercile categories of dietary diversity, such as low (HDDS < 4.5), medium (HDDS = 4.5–6.7) and high (HDDS > 6.7), and it is mapped spatially using the average household HDDS value of sub-district (Upazila) levels of Bangladesh [1,25]

#### 2.3.2. Percentage of Food Energy from Staples (PFES)

A total of 125 food items consumption data were given by metric units (gram) that were converted into total energy (Kcal) using the Bangladesh food composition table [26]. The total food intake energy was divided by 14 to estimate household members' apparent daily energy intake. The percentage of food energy consumed from staples (i.e., cereals, roots and tubers) per household was also analyzed. The formula is given below:

$$\text{PFES} = \frac{\text{Total energy from staples (Kcal)}}{\text{Total energy from all foods (Kcal)}} \times 100$$

A figure of over 75% of total food energy from staples (PFES) indicates poor diet quality [1]. Based on the PFES, household food consumption quality was divided into four groups in this study, such as very good, good, poor and very poor [1]. The spatial

distribution of this indicator was mapped to observe the level of food consumed from staples in sub-district levels in Bangladesh.

The economic vulnerability to food per household was estimated by the percentage of expenditure on food (PEF) in Bangladesh [1].

### 2.3.3. Percentage of Expenditure on Food (PEF)

To measure PEF, the 14-day recall period food consumption was extrapolated into the monthly expenditure of each household for food. On the other hand, the non-food consumption items were converted into monthly expenditure in the HIES data. So, the percentages of food-related expenditure in terms of the total expenditure of each household were estimated [1]. The formula is given below:

$$\text{PEF} = \frac{\text{Total expenditure on food}}{\text{Total expenditure}} \times 100$$

Later on, the results were illustrated at the rural, urban and city corporation levels. Based on the PEF, the households were divided into four groups of economic vulnerability to food insecurity, such as very high, high, medium and low. Furthermore, the average percentages of expenditure on food per sub-district (Upazila) household were analyzed spatially to illustrate the spatial distribution scenarios in Bangladesh.

### 2.3.4. Characteristics of Household

Household characteristics were also observed to identify factors, such as religion, gender, educational qualification of the household head, number of earning members, variability in the settlement, region of the households, influencing diet quality or risk of food insecurity. Gender and number of earners were subdivided into two categories: (Gender: Male/Female; Number of earners: One earner/more than one earner). Settlement of the households was divided into three categories: Rural, Urban and City corporations. Religion was sub-categorized into two levels (Islam and Others) from the status of head member of the household. The region of the households denoted divisional location (Barishal, Chittagong, Dhaka, Rajshahi, Rangpur, Sylhet, Mymensingh, Khulna) of the households.

Independent variables of this study are sociodemographic conditions (i.e., income, age, gender, education of household head, religion, number of earners and location) of the surveyed households.

### 2.4. Data Analysis

Descriptive, inferential and regression analyses were performed in this study. As the number of households and total food components of the surveyed dataset in HIES 2016 was huge, we applied the RStudio package to retrieve the targeted outcomes from the dataset. Later, the retrieved dataset was synthesized using the STATA software to achieve the targeted HDDS, PFES and PEF of household level in Bangladesh. Then, the synthesized outcome of these indicators was analyzed statistically correlating with sociodemographic factors using the ANOVA test. Latest Bangladesh shapefiles were used to map the spatial difference of diet quality as well as economic vulnerability to food insecurity with the sub-district level's weighted average value using ArcGIS 10.1.

## 3. Results

### 3.1. Demographic Characteristics

The majority (69.7%) of the households in the study resided in rural areas. Only 12.7% of the households were headed by female householders. In the study population, 38.3% of the householders were illiterate. About two-third (65.6%) of the households had one earner. Most (87.0%) of the householders were Muslim. A slightly greater share (20.3%) of the households were from the Dhaka division (Table 1).

**Table 1.** Demographic distribution of the observation households by spatial variables (*n* = 46,076).

| Variables | Categories | *n* (%) |
|---|---|---|
| Settlements of households | City corporation | 2120 (4.6) |
| | Rural | 32,096 (69.7) |
| | Urban | 11,860 (25.7) |
| Gender of household head | Female-headed | 5802 (12.7) |
| | Male-headed | 40,066 (87.4) |
| Education of household head | Graduate | 2680 (5.8) |
| | High school | 12,015 (26.2) |
| | Post-graduate | 563 (1.2) |
| | No education | 17,577 (38.3) |
| | Primary | 13,033 (28.4) |
| Level of literacy | Illiterate | 17,577 (38.3) |
| | Literate | 28,291 (61.7) |
| Earner group | More than one earner | 15,850 (34.4) |
| | One earner | 30,218 (65.6) |
| Religion | Islam | 39,925 (87.0) |
| | Others | 5943 (13.0) |
| Region by division | Barisal | 4320 (9.4) |
| | Chittagong | 7916 (17.2) |
| | Dhaka | 9360 (20.3) |
| | Khulna | 7200 (15.6) |
| | Mymensingh | 2880 (6.3) |
| | Rajshahi | 5760 (12.5) |
| | Rangpur | 5760 (12.5) |
| | Sylhet | 2880 (6.3) |

Source: Datasets of HIES 2016 (BBS, 2019).

*3.2. Spatial Difference in Diet Quality and Economic Vulnerability to Food Insecurity*

3.2.1. Household Dietary Diversity Score (HDDS)

The analyzed outcome of HDDS delineated that the average national diet diversity was about 6.3 in Bangladesh during the study period (Figure 1). Divisionally, the range of HDDS was from 5.6 to 7.2. The highest average household diet diversity prevailed in Chittagong and the lowest in the Rangpur division (Figure 1). When HDDS was analyzed against the settlement, low dietary diversity was observed in rural areas (Figure 1). The spatial distribution of HDDS showed that the north, northwest and southeast regions were experiencing mostly low diet diversity (Figure 1).

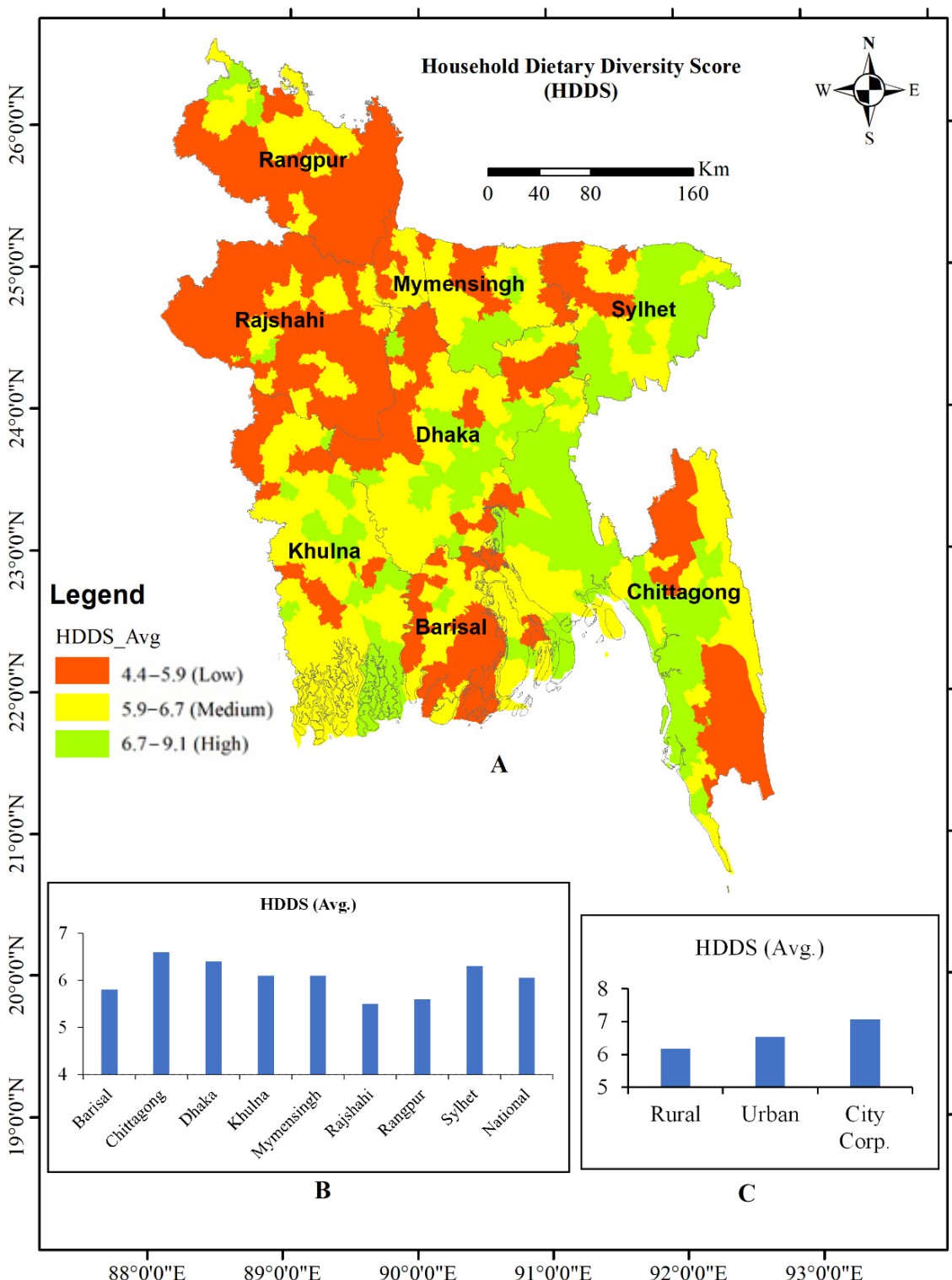

**Figure 1.** Spatial distribution of average HDDS per sub-district (Upazila) level in Bangladesh (**A**); Average HDDS score per household of divisions and national levels in Bangladesh (**B**); and Average HDDS per household of area type in Bangladesh (**C**).

Figure 2 exhibits the average household dietary diversity per food consumption decile. The results showed a positive relationship between the average dietary diversity and consumption per decile (Figure 2). The lowest deciles reported about 5.6 in HDDS, suggesting that the diet of the poor households was barely nutritionally diverse. On the

other hand, households of the higher decile had an HDDS of 7.0 or more, indicating better nutritional diversity and adequacy in those households.

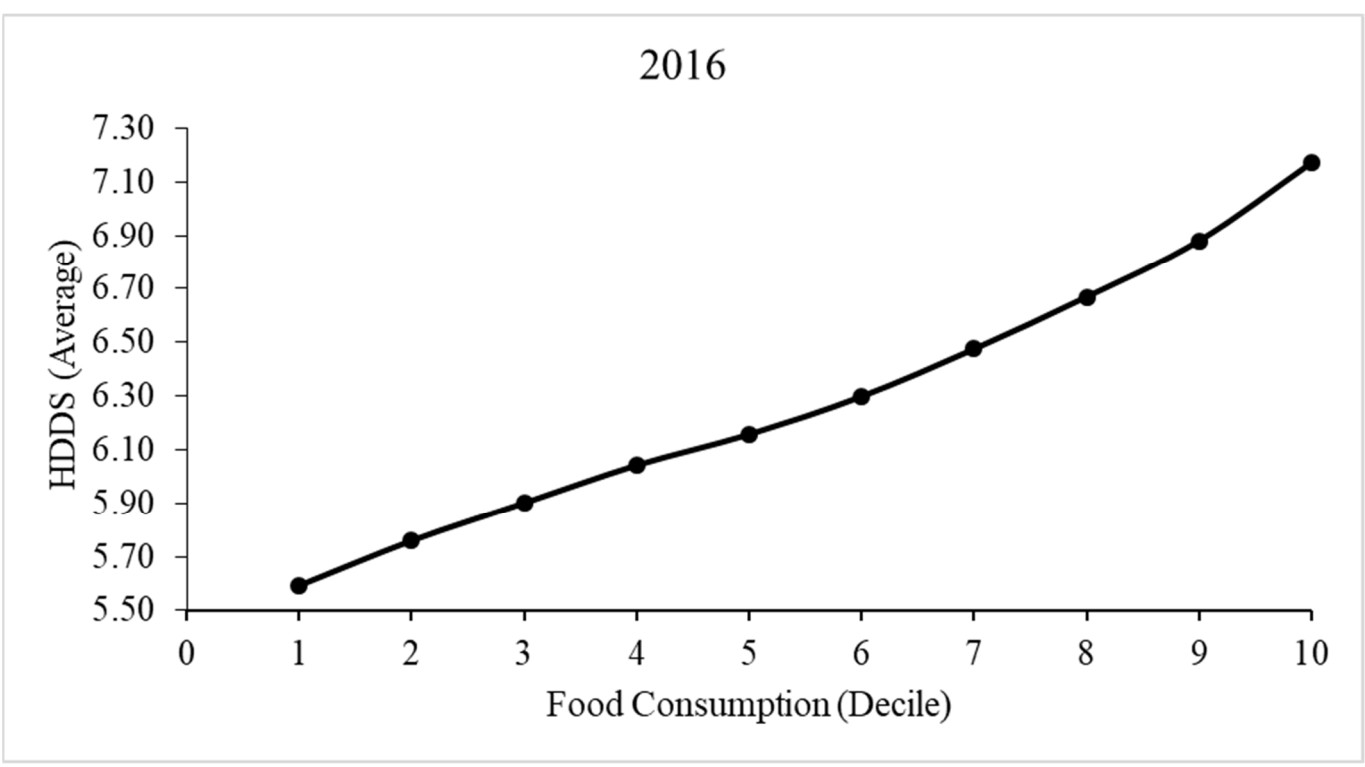

**Figure 2.** Average HDDS by food consumption decile.

3.2.2. Percentage of Food Energy from Staples (PFES)

The percentage of the energy acquired from the staples per day by each household was derived in this study. The national average energy consumption from the staples per household per day was about 70.4% (Figure 3). Similarly to HDDS, poor-quality diet in terms of high PFES also prevailed in rural areas (71.4%) among different kinds of settlements and in Rangpur division (76.9%) among all divisions (Figures 3 and 4). Furthermore, like HDDS, north, northwest and southeast regions illustrated poor diet quality, with more dependence on staples as a source of energy. Besides, the central region (regions around the capital, Dhaka) showed better diet quality with lesser dependence on staples.

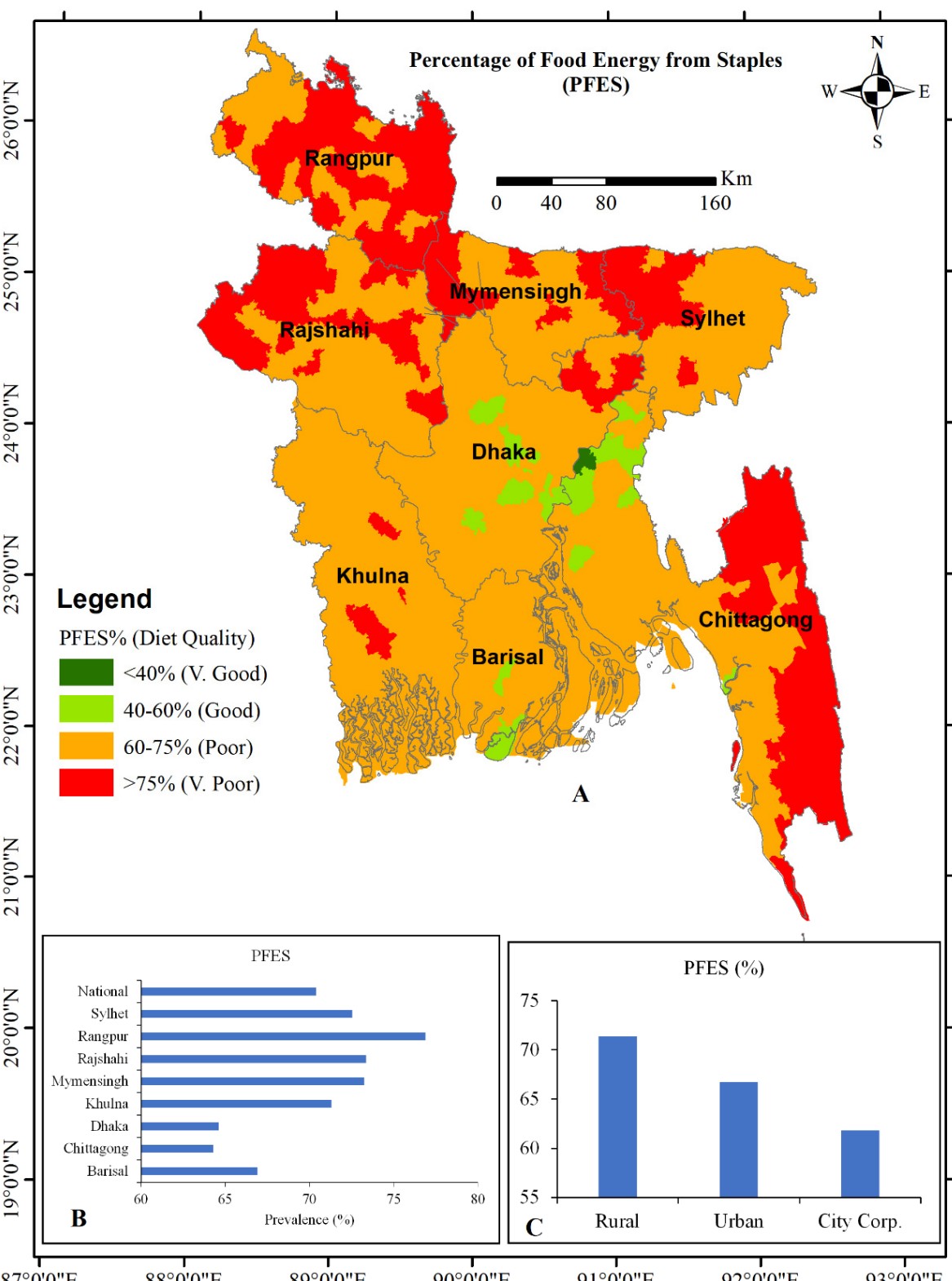

**Figure 3.** Spatial distribution of diet quality in sub-district (Upazila) level, derived using the average PFES (**A**); Average household percentage of food energy from staples in divisions and national level (weightage adjusted) of Bangladesh (**B**); and Percentage of food energy from staples in area type per household (avg.) (**C**).

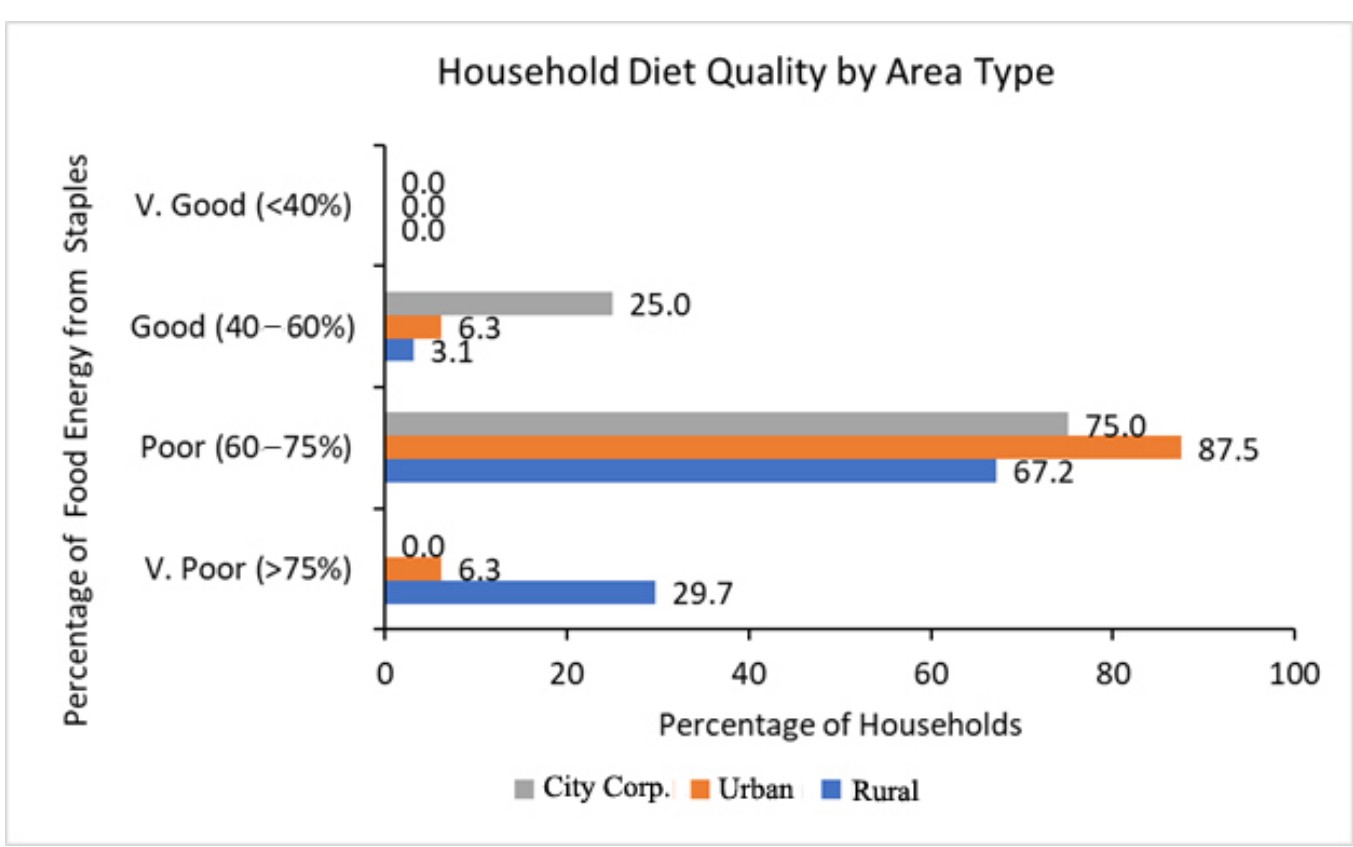

**Figure 4.** Diet quality in households by area type, derived using the PFES (district level weightage adjusted).

3.2.3. Percentage of Expenditure on Food (PEF), i.e., Current Economic Vulnerability

In this study, the percentage of household expenditure devoted to food was calculated using monthly total consumption expenditure. In 2016, the average PEF per family per day was 54.3% (Figure 5). With the highest PEF (57.4%), the Mymensingh division exhibited the highest vulnerability to food insecurity, and Khulna division had the lowest PEF (50.5%) (Figure 5). In terms of settlements, rural areas had comparatively high PEF (55.2%) (Figure 5). At the sub-district level, a medium level of vulnerability prevailed throughout the country (Figure 6). However, the central parts of the country again seemed to be at the lowest risk of vulnerability (Figure 5). At district levels, Sherpur and Feni, and at sub-district levels, Paba Upazila (sub-district of Rajshahi) and Sutrapur Upazila (sub-district of Dhaka), were found to have the highest and the lowest levels of vulnerability to food security, respectively.

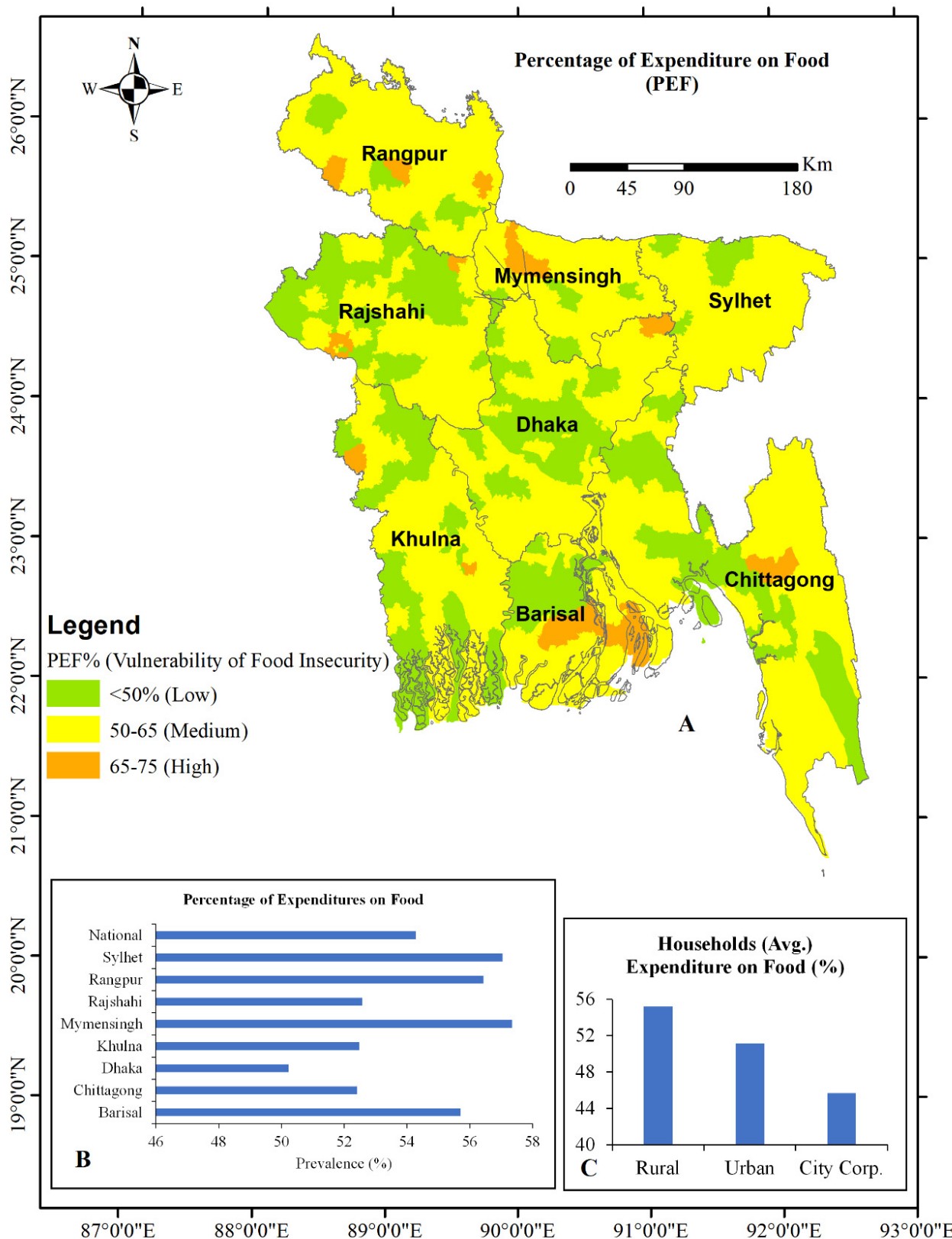

**Figure 5.** Spatial distribution of vulnerability to food insecurity in sub-district level, derived using the average PEF (**A**); Prevalence of PEF per household (avg.) in divisions and national level of Bangladesh (**B**); and Household (avg.) expenditure on food by area type (**C**).

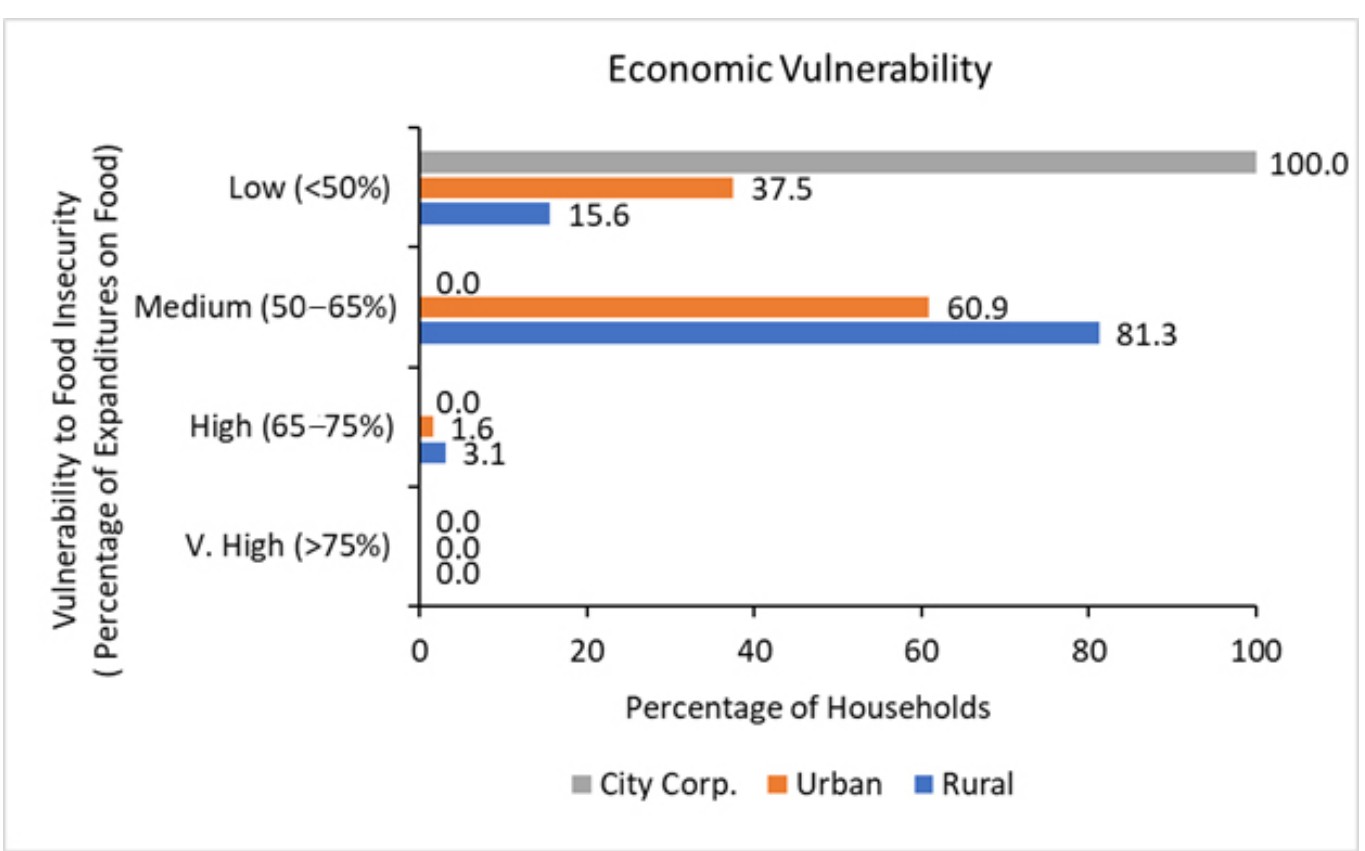

**Figure 6.** Economic vulnerability to food insecurity at the household level (district weightage adjusted), derived from the percentage of total expenditure on food.

3.2.4. Relationship between Sociodemographic Characteristics and Diet Qualities of Households

Diet quality parameters (i.e., HDDS, PFES and PEF) of sample households were correlated with characteristics of nine sociodemographic determinants (Table 2). The results stipulated significant relationships between sociodemographic determinants and dietary quality parameters in this study ($p < 0.05$) (Table 2). Female-headed households seemed to have significantly high HDDS (i.e., HDDS: 6.43 vs. 6.27; $p < 0.001$) and low PFES and PEF. The literacy level and education of the household head, household income and increased number of earning members also reciprocated in significantly higher diversity scores ($p < 0.001$) and low PFES and PFE. In Muslim households, HDDS was slightly higher (i.e., 6.30 vs. 6.23; $p < 0.001$) and vulnerability to food insecurity was lower. Middle-aged household heads showed a higher quality of diet (both in terms of high HDDS and low PFES) than others. Diet quality and EVFI differed significantly ($p < 0.001$) across the divisions. Households in rural areas had significantly lower HDDS and higher PFES and PEF.

**Table 2.** Sociodemographic characteristics of households with average household dietary diversity score, food energy percentage from staples and total expenditure percentage on food (*n* = 46,076).

| Determinants | Frequency, % | | HDDS | | % of Food Energy from Staples (PFES) | | % of Expenditure on Food (PEF) | |
|---|---|---|---|---|---|---|---|---|
| | *n* | % | Mean | *p*-Value | % | *p*-Value | % | *p*-Value |
| (i) Gender of household head | | | | | | | | |
| Male | 40,066 | 87.4 | 6.27 | <0.001 | 70.4 | <0.001 | 54.1 | <0.001 |
| Female | 5802 | 12.7 | 6.44 | | 66.1 | | 52.2 | |
| (ii) Education of household head | | | | | | | | |
| Post-graduate | 563 | 1.2 | 6.72 | | 62.0 | | 42.5 | |
| Graduate | 2680 | 5.8 | 6.76 | | 62.1 | | 42.8 | |
| High school | 12,015 | 26.2 | 6.43 | <0.001 | 68.8 | <0.001 | 50.0 | <0.001 |
| Primary | 13,033 | 28.4 | 6.31 | | 71.9 | | 54.9 | |
| No education | 17,577 | 38.3 | 6.18 | | 72.8 | | 56.6 | |
| (iii) Literacy of household head | | | | | | | | |
| Literate | 28,291 | 61.7 | 6.39 | <0.001 | 69.9 | <0.001 | 51.9 | <0.001 |
| Illiterate | 17,577 | 38.3 | 6.18 | | 72.8 | | 56.6 | |
| (iv) Earner of household | | | | | | | | |
| More than one | 15,850 | 34.4 | 6.34 | <0.001 | 70.1 | <0.001 | 53.0 | <0.001 |
| One | 30,218 | 65.6 | 6.27 | | 69.7 | | 54.2 | |
| (v) Religion of household head | | | | | | | | |
| Islam | 39,925 | 87.0 | 6.30 | <0.001 | 69.4 | <0.001 | 53.7 | <0.001 |
| Others | 5943 | 13.0 | 6.24 | | 72.8 | | 54.4 | |
| (vi) Age of household head | | | | | | | | |
| Young-age adult (18–35) | 14,106 | 30.6 | 6.29 | | 68.8 | | 55.0 | |
| Middle-age adult (36–55) | 21,508 | 46.9 | 6.30 | <0.001 | 70.7 | <0.001 | 52.9 | <0.001 |
| Older adult (>55) | 10,256 | 22.4 | 6.27 | | 69.7 | | 54.1 | |
| (vii) Region by division | | | | | | | | |
| Barisal | 4320 | 9.4 | 6.0 | | 67.3 | | 55.1 | |
| Chittagong | 7916 | 17.2 | 6.84 | | 67.8 | | 53.8 | |
| Dhaka | 9360 | 20.3 | 6.54 | | 64.6 | | 52.3 | |
| Khulna | 7200 | 15.6 | 6.20 | <0.001 | 70.9 | <0.001 | 52.3 | <0.001 |
| Mymensingh | 2880 | 6.3 | 6.33 | | 72.9 | | 58.2 | |
| Rajshahi | 5760 | 12.5 | 5.73 | | 73.0 | | 51.7 | |
| Rangpur | 5760 | 12.5 | 5.78 | | 75.6 | | 55.8 | |
| Sylhet | 2880 | 6.3 | 6.75 | | 72.6 | | 56.6 | |
| (viii) Monthly income (Quintile) | | | | | | | | |
| Lowest | 7723 | 16.9 | 6.07 | | 71.8 | | 56.3 | |
| Low | 8552 | 18.7 | 6.06 | | 74.1 | | 56.4 | |
| Medium | 9099 | 19.9 | 6.31 | <0.001 | 71.9 | <0.001 | 55.6 | <0.001 |
| High | 9634 | 21.1 | 6.40 | | 70.5 | | 53.2 | |
| Highest | 10,703 | 23.4 | 6.60 | | 67.6 | | 48.5 | |
| (ix) Area type | | | | | | | | |
| Rural | 32,096 | 69.7 | 6.13 | | 71.9 | | 55.6 | |
| Urban | 11,860 | 25.7 | 6.61 | <0.001 | 65.8 | <0.001 | 50.6 | <0.001 |
| City corporation | 2120 | 4.6 | 7.10 | | 61.7 | | 45.7 | |

*p*-value < 0.05 was considered significant by ANOVA test.

## 4. Discussion

The spatial disparity has the potential to disrupt sustainable food security systems, so the primary objective of the research was to examine the existing regional differences in diet quality, economic vulnerability to food insecurity (EVFI) and factors associated with them. While there is burgeoning research on the impact of geographical location on health outcomes, in Bangladesh, no examination of regional differences in diet quality and EVFI has been conducted so far. Our findings suggest that dietary quality and economic vulnerability to food insecurity are spatially dependent, indicating poor-quality diet in rural and peripheral areas of the country.

Regional and geographical differences in diet quality and EVFI were also prominent in the study. In particular, the highest level of food insecurity in Bangladesh was not observed. This might be due to the weighting of the sampled households' food expenditure figures by the district level of weightage factor. Through the lens of settlement, city dwellers had better diet quality both in terms of HDDS and PFES (Figures 3 and 4). Moreover, a significantly low percentage of city dwellers were financially prone to food insecurity. This was in parallel with earlier study findings among older adults [4]. When the location of the households by division was considered, significant quality and vulnerability variations were observed. The central parts of the country seemed to be at lower risk of food insecurity and enjoyed better diet quality, while the peripheral parts (north, northwest and southeast) of the country, which consist of the hilly and coastal areas and disaster-prone areas, seemed to have a higher economic vulnerability to food insecurity. This was also mirrored in their poor food intake quality. This finding is also in line with an earlier study of Bangladesh, where the authors reported that the north and south parts are more vulnerable to food insecurity [20].

Access to different food groups varies in terms of both physical and economic accessibility. A study by Park et al. reported that the socioeconomic conditions of regions might be the underlying causes for such association with diverse food intake [4]. Bangladesh is largely dependent on agriculture for food. Although both per capita food intake and productivity have increased over time in Bangladesh, a large portion of the population is food insecure due to their low purchasing power to access an adequate diet [27]. Previous studies have also reported the dependence of rural regions on starch/staple-concentrated diet and their poor dietary diversity [28]. Agricultural performance, rainfall frequency, land degradation, climate change, natural disasters, including drought and flood, also vary from region to region [9]. These factors create a difference in vulnerability to food insecurity and unequal diet quality throughout the country [29].

The diet quality parameters (HDDS, PFES and PEF) used in this study were associated with gender, education, literacy, religion, age, division, number of earners, monthly income level and residency type of the households. Household head (female), educational qualification (literate with a higher level of education), age (middle aged), income (highest quintile), number of earning members (more than one), religion (Islam) and area type (city) reciprocated in better diet quality. There are significant influences of sociodemographic and lifestyle factors on the diet quality of households observed in various studies [3,4,8,20,22,28,30–32]. It has been observed that female-headed households had significantly better diversity than male-headed households. In female-headed homes, previous studies reveal that women's decisions and incomes are often directed toward the welfare of the entire household's diet quality, nutrition and health [33,34]. However, an earlier study reported otherwise and pointed out women's immobility to lower diet quality in women-headed households [22]. Better literacy level, household income and several bread earners were found to positively influence the household dietary diversity. The findings are consistent with the previous studies [32,33]. A previous study suggests that having an educated female household member can increase dietary diversity by about 5–6% and can contribute to raising the HDDS by around 0.6 points [22]. Moreover, educated households are expected to have a greater capacity to perceive and practice general dietary guidelines and adapt nutritional advice and cooking methods, which in turn improves their

dietary diversity [22,35]. The association between income, several earners and diet quality might be attributed to the high price of healthy foods, which limits their accessibility to the low-income households [35].

This study implies that diet-related inequalities are both associated with the microsystem level, such as socioeconomic status, and the macrosystem level, such as the regional level. Varied food environments and diverse regional foodways impact food availability, accessibility, and thus, dietary choices [12]. Although the Bangladesh government has made immense strides in economic growth, health and food security system, there is a lack of understanding of regional differences regarding diet quality, as well as sustainable food security system. With proper measures, a sustainable food security system and adequate diet quality can be attained within various regions in the country, and media talks can help regarding this issue [36].

This research has several strengths. The random and multi-stage sampling scheme and a representative sample size facilitated the generalization of the results to the entire Bangladeshi population. No earlier study in Bangladesh attempted to identify spatial inequality at the sub-national level in terms of diet quality and vulnerability to food insecurity. Furthermore, the majority of putative determining factors related to food insecurity and diet quality were included in this analysis. However, this study had some limitations that need to be addressed in future research. Seasonal variation in diet quality and economic vulnerability to food insecurity were not analyzed. The results indicated inequality in diet quality only at the household level; individual diet quality is yet to be assessed. As the geographical disparity between diet quality and vulnerability to food insecurity was observed only at a specific time point, the causality and the trend could not be determined.

## 5. Conclusions

The spatial distribution of diet quality gaps at the household level was delineated with the sociodemographic characteristics using the HIES data of 2016 in Bangladesh. Diet quality deteriorated from the central to the peripheral parts of the country due to the disproportionate availability of resources and food sources in those parts of the country. Diet quality analysis indicated that the north, northwest and southeast regions of Bangladesh were mostly low on diet quality. Most of the sub-districts/districts showed a medium level of food insecurity vulnerability, whereas countryside areas showed a higher level based on the percentage of expenditure on food. The relationship between sociodemographic variables and diet quality was shown to be substantial, with better diet qualities reported in females, middle aged, Muslim and higher-educated household heads, as well as higher-income households and those living in cities. Proper understanding of the underlying factors causing disparity in dietary quality and economic vulnerability to food security can assist policymakers in undertaking proper interventions and establishing equality and food sustainability in these sectors. The occurrence of regional food quality variations underscores the importance of sub-district-level interventions to enhance access to nutritious non-staple foods. The study also suggests establishing women empowerment, nutrition and general education, and income-generation activities in targeted areas, as these characteristics were linked to better diet quality.

**Author Contributions:** Conceptualization, M.M.P.M., M.A. and M.R.A.; Data curation, M.M.P.M.; Formal analysis, M.M.P.M. and M.R.A.; Methodology, M.M.P.M., S.S. and M.R.A.; Supervision, M.R.A. and K.I.; Writing—original draft, M.M.P.M.; Writing—review & editing, M.M.P.M., M.R.A., S.S., M.A. and K.I. All authors have read and agreed to the published version of the manuscript.

**Funding:** The APC was funded by University of Dhaka.

**Data Availability Statement:** The datasets for this study are available from Bangladesh Bureau of Statistics (BBS) (http://www.bbs.gov.bd/) upon purchase by the researchers.

**Acknowledgments:** The authors gratefully acknowledge the financial support from the University of Dhaka and Bangladesh, and the Bangladesh Bureau of Statistics (BBS) for providing the HIES 2016 data for this study.

**Conflicts of Interest:** The authors declare no conflict of interest.

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
