# Peer review of "Spatial Differences in Diet Quality and Economic Vulnerability to Food Insecurity in Bangladesh: Results from the 2016 Household Income and Expenditure Survey"

_sustainability, doi:10.3390/su14095643_

Round 1
Reviewer 1 Report
The text is interesting and has a methodological component that can be used in spaces other than the one targeted.
The theoretical part (Introduction) covers a number of relevant concepts, but I would recommend that two more sections or at least paragraphs be inserted:
One should talk about what other studies have been done on diet quality and food security within other societies where, for cultural reasons, or because of differences rural-urban, or because of the economic and social inequality, there are big differences from one region to another, or from one social group to another. Surely there are studies of this type for other countries as well. Because here the problem is not only the quality of food or food insecurity, but also the differences from one region to another or from one social category to another, the inequalities from within the country. It can be studied in this regard, for example, the following article: Socioeconomic inequalities in food insecurity and malnutrition among under-five children: within and between-group inequalities in Zimbabwe | BMC Public Health | Full Text (biomedcentral.com).
A second section, mandatory in the world in which we live (in which the media and the interconnectivity are present almost everywhere) should refer, at least in one/two paragraphs, to the issue of food sustainability and how it appears in the public discourse. And how do people relate to it? What's “the talk” about food sustainability, food security in Bangladesh? Especially in the context of these regional differences from within. That would help us get a better idea of the whole landscape of the problem, and help us understand the text further.
In addition, the awareness of the discourse on food security, sustainability, etc. can be an ingredient in promoting solutions for vulnerable areas or vulnerable people. Especially since the results of your study suggest that social factors are extremely important in reducing food vulnerability. In this regard, the following article from Sustainability can be studied: Sustainability | Free Full-Text | Talking about Sustainability: How the Media Construct the Public's Understanding of Sustainable Food in Romania (mdpi.com)
Author Response
File attached.

Reviewer 2 Report
In the article: “Spatial Differences in Diet Quality and Economic Vulnerability to Food Insecurity in Bangladesh: Results from the 2016 Household Income and Expenditure Survey” The Authors explored the spatial differences of diet quality and economic vulnerability of food insecurity with the association of sociodemographic characteristics at the household level in Bangladesh.
The article contains all the parts that should be found in a scientific article. However, I have some reservations about the content presented (see comments below). There is also some doubt about the validity of the research conducted, based on data from 2016 (here also see the comments below).
My main objections to the article, requiring clarification and possible improvement:
- 1 The authors write in lines 10-11: "This study was a secondary analysis of Household income and expenditure survey (HIES) data of 2016." Nowhere in the article did I find an explanation of what is meant by this statement. Was there a new analysis based on 2016 data? If so, it is difficult to talk about the validity of this survey here. Was the study repeated and the results from 2016 compared? It is not written in the article, but then why are there statements like: lines 15-16: "The Study findings showed that quality of diet degraded and EVFI upgraded from urban to rural and central to peripheral parts of the country.", or lines 299-300: "Diet quality degraded from central to peripheral parts of the country". If there were no repeat surveys where did the conclusions about changes come from?
- What in the article is the Authors' own research results and what was carried over from the 2016 study? e.g. Table 1 presented in the Results section - are these the Authors' research results or data from the 2016 study? No source under the table!
- Chapter Introduction needs to be expanded. Please indicate in a broader way the background of the study, the results of other similar studies conducted in other countries. I would suggest referring to more recent studies. Most of the cited publications are not from recent years.
- Lines 149-150: you have managed to fit your research methodology into two! lines. Please strongly elaborate and describe this point, justifying the choice of methods used based on the literature.
- In the discussion of the results, please refer to studies of this type conducted by other authors in other countries and compare the results obtained with this.
- for all tables and figures please add the source! What is the result of the Authors' analysis and what is just a representation of data from 2016.
- Please comply in your article with MDPI editing requirements (no chapter numbering, poor format of references, paragraphs between lines, final reference list, etc.).
- Typos, e.g., Line 76: “Matetials”
The article needs to be rebuilt and improved.
Author Response
File attached.

Round 2
Reviewer 2 Report
The authors have addressed all my comments. Their explanations and corrections made in the article satisfy me. I accept the article in terms of its merits and recommend it for publication. Only the editing aspect needs improvement (adjustment to the requirements of MDPI, e.g., in the corrected version there are no headings, bad form of citations).